# YG-1 Extract Improves Acute Pulmonary Inflammation by Inducing Bronchodilation and Inhibiting Inflammatory Cytokines

**DOI:** 10.3390/nu13103414

**Published:** 2021-09-28

**Authors:** Hye-Yoom Kim, Jung-Joo Yoon, Dae-Sung Kim, Dae-Gill Kang, Ho-Sub Lee

**Affiliations:** 1Hanbang Cardio-Renal Research Center & Professional Graduate School of Oriental Medicine, Wonkwang University, Iksan 54538, Korea; hyeyoomc@naver.com (H.-Y.K.); mora16@naver.com (J.-J.Y.); dgkang@wku.ac.kr (D.-G.K.); 2Hanpoong Pharm and Foods Co., Ltd., Wanju 55316, Korea; kimezz@naver.com

**Keywords:** YG-1 extract, bronchodilation, fine particulate matter (PM2.5), acute lung injury, airway inflammation

## Abstract

YG-1 extract used in this study is a mixture of *Lonicera japonica*, *Arctic Fructus*, and *Scutellariae Radix*. The present study was designed to investigate the effect of YG-1 extract on bronchodilatation (ex vivo) and acute bronchial and pulmonary inflammation relief (in vivo). Ex vivo: The bronchodilation reaction was confirmed by treatment with YG-1 concentration-accumulation (0.01, 0.03, 0.1, 0.3, and 1 mg/mL) in the bronchial tissue ring pre-contracted by acetylcholine (10 μM). As a result, YG-1 extract is considered to affect bronchodilation by increased cyclic adenosine monophosphate, cAMP) levels through the β2-adrenergic receptor. In vivo: experiments were performed in C57BL/6 mice were divided into the following groups: control group; PM2.5 (fine particulate matter)-exposed group (PM2.5, 200 μg/kg/mL saline); and PM2.5-exposed + YG-1 extract (200 mg/kg/day) group. The PM2.5 (200 μg/kg/mL saline) was exposed for 1 h for 5 days using an ultrasonic nebulizer aerosol chamber to instill fine dust in the bronchi and lungs, thereby inducing acute lung and bronchial inflammation. From two days before PM2.5 exposure, YG-1 extract (200 mg/kg/day) was administered orally for 7 days. The PM2.5 exposure was involved in airway remodeling and inflammation, suggesting that YG-1 treatment improves acute bronchial and pulmonary inflammation by inhibiting the inflammatory cytokines (NLRP3/caspase-1 pathway). The application of YG-1 extract with broncho-dilating effect to acute bronchial and pulmonary inflammation animal models has great significance in developing therapeutic agents for respiratory diseases. Therefore, these results can provide essential data for the development of novel respiratory symptom relievers. Our study provides strong evidence that YG-1 extracts reduce the prevalence of respiratory symptoms and the incidence of non-specific lung diseases and improve bronchial and lung function.

## 1. Introduction

Bronchodilators are important drugs in the treatment of asthma, acute, and chronic obstructive airway disease. Beta agonists, theophylline and antimuscarinic drugs are the main drugs currently used [1]. These drugs are known to directly affect airway smooth muscle and cause bronchodilation [2]. Abnormal state of airway smooth muscle cells is involved in airway remodeling [3]. Excessive exposure to fine particulate matter (PM2.5) is gradually absorbed into the bronchi and lungs and progresses to acute respiratory distress syndrome, eventually requiring bronchodilation for respiration [4]. Therefore, it is meaningful to apply natural products with bronchodilating effect to animal models of acute bronchial and lung inflammation. Particulate matter (PM) is one of the various artificial pollutants worldwide and has recently received much attention due to its biohazard effects. PMs are classified into two groups, PM10 and PM2.5, according to their size. PM10 refers to particulate matter less than ten μm in diameter, and PM2.5 refers to particles less than 2.5 μm in diameter [4,5]. Respiratory symptoms and diseases are becoming more and more serious due to air pollution and environmental changes caused by rapid industrialization [5]. Respiratory symptoms caused by air pollution are caused by an inflammatory reaction in the bronchi with stimuli such as fine dust, which causes or worsens acute sepsis, asthma, chronic bronchitis, and airway obstruction [6,7]. Recently, exposure to PM2.5 has been identified as a major risk factor for respiratory diseases [8,9]. PM2.5 not only causes respiratory dysfunction (cough and wheezing etc.) but also worsens the condition, increasing morbidity and mortality [10,11]. Also, it was reported that PM2.5 induces airway inflammation in mice and nasal inoculation of enriched PM2.5 induces an inflammatory airway response [12]. According to various studies, short-term exposure to PM2.5 in mice is known to induce acute lung inflammation [13].

Currently, the treatment of respiratory diseases is dependent on the use of drugs such as bronchodilators and anti-inflammatory drugs, but alternative medicine using natural products with few side effects is needed [14]. Among natural products, there are many ingredients known to have antitussive expectorant effects that can treat respiratory symptoms and diseases. It has been reported that these natural products have clinical effects in relieving respiratory symptoms when administered alone or in combination [15]. Mixtures of natural products have been used for the treatment of various diseases since ancient times, and their physiologically active efficacy has been verified based on long-term experience and is widely used because there are few side effects [16]. The YG-1 extract used in this study is a mixture of *Lonicera japonica, Arctii Fructus,* and *Scutellariae Radix* (Table 1). *Lonicera japonica*, which accounts for a significant proportion of the YG-1 mixed extract, is known to have antipyretic, detoxifying, and sweating effects [17], and *Arctii Fructus* is used to relieve fever and sore throat [18]. In addition, *Scutellaria Radix* has anti-inflammatory, antipyretic, diuretic, and blood pressure lowering effects, and is currently used for chronic bronchitis, infectious hepatitis, and hypertension [19]. In this study, we assess the bronchodilatation (ex vivo) and acute bronchial and lung inflammation relief effects (in vivo) of YG-1 extract, a mixture of natural products (*Lonicera japonica*, *Arctii Fructus*, and *Scutellariae Radix*) widely used as antitussive expectorants in folk remedies.

## 2. Materials & Methods

### 2.1. Preparation of YG-1 Extract

The YG-1 extract was a mixed extract containing *Lonicera japonica, Arctii Fructus,* and *Scutellariae Radix,* which was provided by Hanpoong (Hanpoong Pharm and Foods Co., Ltd., Wanju, Korea). Lonicera japonica, and Arctii Fructus were each added at a ratio of 3:1, and 20 times 30% alcohol was added, followed by extraction twice at 85–95 °C for 3 h each. After filtration, the extract was concentrated under reduced pressure at 60 °C or less and dried to prepare YG-A (yield 14%). *Scutellariae Radix* was extracted twice for 3 h at 85–95 °C by adding 20 times 30% alcohol, and concentrated and dried to prepare YG-B (yield 45.61%). YG-1 was prepared by mixing dried YG-A and YG-B in a ratio of 2:3 (Table 1) according to the ratio previously used in the study [20].

### 2.2. HPLC Analysis of YG-1

Seven reference standard components, loganin, loganic acid, sweroside, arctiin, baicalin, baicalein, and wogonin were purchased from ChemFaces (ChemFaces Biochemical, Wuhan, China), respecitively. Chemical structures of these reference standard components are shown in Figure 1. HPLC analysis for the comparison of the 7 marker components in YG-1 extract was performed using HPLC instrument (I-series, LC-2030C, Shimadzu, Kyoto, Japan), PDA detector (Shimadzu, Kyoto, Japan) and LC Solution software (Version 1.24, SP1, Shimadzu, Kyoto, Japan). Analysis of loganin, loganic acid, sweroside, arctiin, baicalin, baicalein, wogonin were performed using a Capcell Pak HPLC Columns (250 × 4.6 mm I.D, C18 UG120 column, 5 μm, Osaka soda, Japan).

### 2.3. Isolation of Bronchial Tissue and Measurement of Bronchodilation (Ex Vivo)

After the head of a healthy male Sprague-Dawley (weighing approximately 250–300 g) was dislocated, the rib cage was excised, and the bronchi were isolated. The rapidly isolated bronchial were saturated with mixed gas (95% O_2_ and 5% CO_2_) in a Krebs solution (118 mM NaCl; 1.5 mM CaCl_2_; 4.7 mM KCl; 25 mM NaHCO_3_; 10 mM glucose; 1.1 mM MgSO_4_; and 1.2 mM KH_2_PO_4_; pH 7.4 with ice-cold), remove the surrounding fat and connective bronchial tissue, and then cut into sections with a length of about 3–4 mm. At this time, be careful not to damage the bronchial smooth muscle. 5 mL of Krebs solution was placed in the chamber and maintained at 37 °C with the mixed gas. The detached bronchial ring was pulled up to a force of 1.8 g and equilibrated for 60 min. Isometric tension changes were recorded via a connected transducer (Grass FT 03, Grass Instrument Co., Quincy, MA, USA) and a Grass Polygraph recording system (Model 7E, Grass Instrument Co., West Warwick, RI, USA). The bronchodilation reaction was confirmed by treatment with YG-1 concentration-dependently (0.01, 0.03, 0.1, 0.3, and 1 mg/mL) in the bronchial tissue ring pre-contracted by acetylcholine (10 μM).

### 2.4. Measurement of cAMP Levels in Bronchial Tissues

After equilibrating the bronchial sections in Krebs solution for 30 min while supplying 95% O_2_ and 5% CO_2_ mixed gas, 3-isobutyl-1-methylxanthine (IBMX, 100 μM) and acetylcholine (100 μM) were added to equilibrate for another 5 min. Each concentration was treated with YG-1 extract (1, 2.5, and 5 mg/mL, respectively) and reacted for 10 min. In addition, KT5720 (100 uM) was pre-treated 20 min before, and YG-1 was treated with 5 mg/mL and reacted. The bronchial tissue was immediately put in liquid nitrogen to stop the reaction, and then stored at −70 °C and used to measure the cAMP concentration. After homogenizing the vascular tissue whose weight was measured in the presence of 0.1M HCl, the supernatant obtained by centrifugation at 13,000 g for 15 min was used with Dirict cAMP ELISA kit (Enzo, ADI-900-066, Biotechnologies Corp & Enzo Life Sciences, New York, NY, USA) was measured.

### 2.5. PM2.5-Induced Acute Lung and Bronchial Inflammation Mouse Model

After acclimatization for 7 days, all mice were randomly divided into 3 groups (*n* = 8 per group). Experiments were performed in C57BL/6 mice were divided into the following groups: control group; PM2.5-exposed group (PM2.5, 200 μg/kg/mL saline); and PM2.5-exposed + YG-1 extract (200 mg/kg/day) group. From two days before PM2.5 exposure, YG-1 extract (200 mg/kg/day) was orally administered for a total of 7 days. The YG-1 extract was exposed to PM 2.5 for 5 days from two days after the start of feeding to induce acute bronchial and lung inflammation and confirm the improvement effect of YG-1. PM2.5 purchased from Sigma Aldrich (NIST1650b, St. Louis, MO, USA) was dissolved in dimethyl sulfoxide (DMSO, 100%) and washed three times with deionized distilled water for treatment, and ultrasonic pulverization was performed for 3 min to minimize agglomeration. PM2.5 (200 μg/kg/mL saline) was exposed for 1 h for 5 days using an ultrasonic nebulizer aerosol chamber (Mass Dosing Chambers, Data Sciences International, Saint Paul, MN, USA) to instill fine dust in the bronchi and lungs, thereby inducing acute lung and bronchial inflammation. Control group received the same amount of saline used as the dosing vehicle. PM2.5 exposure procedures have been referenced based on various studies [21,22]. C57BL/6 mice were exposed to PM2.5 in the awake and uninhibited state and continuously received concentrated ambient air PM2.5 following an in vivo systemic inhalation protocol. The animals in this study were conducted after obtaining approval from the Animal Experiment Ethics Committee of Wonkwang University (ethics review number: WKU20-28).

### 2.6. Histological Analysis

The lung and bronchial tissues isolated from mice in each group were fixed in 10% neutral buffered formalin (10% NBF, HT501128, Merk, Darmstadt, Hessen, Germany) solution for 24 h. After perfusion fixation and paraffin embedding, paraffin blocks were cut into 6–7 μm thick tissue sections using a microtome (Thermo Electron Corporation, Pittsburg, PA, USA) and attached to slides. The lung and bronchial tissue slides were prepared using Periodic Acid Solution (PAS, VB-3005, VitroVivo Biotech, Rockville, MD, USA), Masson’s trichrome (8400, BBC Biochemical, Mt Vernon, WA, USA), and orcerin (ab245881, Abcam, Cambridge, Cambs, UK) stained with a stain kit. Also, the beta-AR, TGF-beta, collagen IV proteins in lung and bronchial tissues were examined with immu nohistochemical (IHC) staining. The lung and bronchial tissue slides were immune stained by mouse and rabbit specific HRP/DAB (ABC) detection IHC kit method (ab6464, Abcam, Cambridge, Cambs, UK). Tissue sections were incubated with primary antibodies of beta 2 Adrenergic Receptor (B2AR, MBS8543138, MyBioSource, San Diego, CA, USA), TGF-β, and collagen IV (Santa Cruz Biotechnology, Santa Cruz, CA, USA). Histopathological comparisons were performed with a microscope slide scanner (MoticEasyScan Pro 1, National Optical & Scientific Instruments, Inc., Schertz, TX, USA).

### 2.7. Western Blot Analysis and Antibodies

The lung and bronchial tissues (30–45 μg protein) were resolved on 10% SDS-PAGE (SDS-polyacrylamide gel electrophoresis) and transferred onto PVDF (polyvinylidene difluoride) western blot membranes. The membranes were washed three times with TBS-T (Tris buffered saline: 150 mM, NaCl; 10 mM, Tris-HCl; and 0.05%, Tween-20) and blocked with 5% BSA (bovine serum albumin) for 2 h. After that, it was washed again 3 times with TBS-T and reacted overnight with appropriate primary antibodies (tumor necrosis factor alpha, TNF-α; interleukin-6, IL-6; interleukin-1β, IL-1β; interleukin-18, IL-18; NOD-like receptor pyrin domain-containning protein 3, NLRP-3, apoptosis-associated speck-like protein containing a C-terminal caspase recruitment domain, ASC; caspase-1) overnight at 4 °C. TNF-α, IL-6, IL-1β, IL-18, NLRP-3, ASC, and caspase-1, and β-actin were purchased from Santa Cruz (Santa Cruz Biotechnology, Dallas, TX, USA). The next day, the membrane was washed three times with TBS-T and reacted with a secondary antibody conjugated to horseradish peroxidase (Bethyl Laboratories, Montgomery, TX, USA) for 2 hr. For the membrane reacted with the secondary antibody, the protein expression level was confirmed using an image analyzer (iBright FL100, Thermo Fisher Scientific, Waltham, MA, USA). The ImageJ program (NIH, Bethesda, MD, USA) was used to quantify protein levels by performing densitometry analysis.

### 2.8. Quantitative Real-Time Reverse Transcription-PCR of Lung and Bronchial Tissues

#### The Real-Time qRT-PCR of Lung and Bronchial Tissues

To confirm the real-time quantitative reverse transcription polymerase chain reaction (qRT-PCR) of lung and bronchial tissues, RNA was extracted from each tissue using Trizol™ Reagent (15596026, ThermoFisher Scientific, Waltham, MA, USA). The cDNAs from lung and bronchial tissues were incubated in SimpliAmp™ Thermal Cycler (A24811, ThermoFisher Scientific, Waltham, MA, USA) at 42 °C for 60 min and 94 °C for 5 min, and synthesized from mRNA through reverse transcription. The real-time qRT-PCR was performed with an initial denaturation step at 95° in a final volume of 20 µL (1 µL of cDNA sample; 1 µL of primer pair each; 8 µL, pure distilled water; 10 µL of SYBR™ Green PCR Master Mix, 4309155, ThermoFisher Scientific, Waltham, MA, USA). Reactions were performed at 95 °C for 10 min using the Step-One™ Real-Time PCR system, followed by 40 repetitions at 95 °C for 15 s and finally 60 °C for 60 s (Applied Biosystems, ThermoFisher Scientific, Waltham, MA, USA). The sequences of primers were as follows: IL-6 (forward, 5′-AACTCCATCTGCCCTTCA-3′; reverse, 5′-CTGTTGTGGGTGGTATCCTC-3′), IL-1β (forward, 5′-TTCAAATCTCACAGCAGCAT-3′; reverse, 5′-CACGGGCAAGACATAGGTAG-3′), NLRP3 (forward, 5′-CTGGAGATCCTAGGTTTCTCTG-3′; reverse, 5′-CAGGATCTCATTCTCTTGGATC-3′), ASC (forward, 5′-CTCTGTATGGCAATGTGCTGAC-3′; reverse, 5′- GAACAAGTTCTTGCAGGTCAG-3′), Caspase 1 (forward, 5′-GAGCTGATGTTGACCTCAGAG-3′; reverse, 5′- CTGTCAGAGAGTCTTGTGCTCTG-3′), TNF-α (forward, 5′-GCCTCTTCTCATTCCTGCTTG-3′; reverse, 5′-CTGATGAGAGGGAGGCCATT-3′), and β-actin (forward, 5-GGAGATTACTGCCCTGGCTCCTAGC-3′; reverse, 5′-GGCCGGACTCATCGTACTCCTGCTT-3′).

### 2.9. Statistical Analyses

All experiments were repeated at least 3 times, and statistically significant differences between group means were determined using Student’s t-test. Results of experiments were expressed as mean ± standard error (S.E.). *p* < 0.05 was considered a statistically significant difference.

## 3. Results

### 3.1. HPLC Chromatograms of Compounds from YG-1 Extract

Figure 1 shows the chromatograms analyzed by high performance liquid chromatography (HPLC) for *Lonicera japonica* (loganin, loganic acid, and sweoside), *Arctii Fructus* (arctiin), and *Scutellariae Radix* (baicalin, baicaein, and wogonin) from YG-1 extract (Figure 1). Chromatograms were detected at 254 nm for loganin, loganic acid, and sweroside, arctiin at 280 nm, and baicalin, baicalein, and wogonin at 277 nm using a photodiode array detector (Figure 2). As a result of analyzied YG-1 extract, Loganin (5.80 ± 0.16 mg/g), Loganic acid (2.38 ± 0.54 mg/g), and Sweoside (3.21 ± 0.07 mg/g) contained in *Ronica japonica*; Arctiin (42.67 ± 0.22 mg/g) contained in *Arctii Fructus*; And Baicalin, Baicaein, and Wogonin contained (sum of 3 compounds: 118.67 ± 2.34 mg/g) in *Scutellariae Radix* could be identified, respectively (Figure 2).

### 3.2. Concentration-Dependent Bronchodilation Effect of YG-1 Extract in Bronchial Smooth Muscle

The contraction of bronchial (tracheal) smooth muscle was induced with acetylcholine 10 μM, and the concentration-dependent bronchial relaxation effect of YG-1 extract was confirmed. As a result, bronchial smooth muscle showed a significant relaxation effect at the 5 mg/mL concentration of YG-1 extract compared to 97.58 ± 11.02% of untreated bronchial smooth muscle (Figure 3A(a). In order to examine whether YG-1 extract affects cAMP production in bronchial tissues, the amount of cAMP production was measured by treatment in a concentration-dependent manner. As a result, it was possible to confirm a significant increase in cAMP production in a concentration-dependent manner compared to the group not treated with the YG-1 extract (Figure 3A(b). In addition, the bronchial rings were pre-treated with YG-1 extract (2.5 or 5 mg/mL concentration) to determine whether contraction by acetylcholine. As a result, the YG-1 extract inhibited acetylcholine-induced contraction in a concentration-dependent manner (Figure 3B(a). Therefore, it is considered that the YG-1 extract has the effect of inhibiting bronchi contraction. Also, it is thought that the YG-1 extract has a broncho-dilating effect and is involved in cAMP production.

### 3.3. Effects of YG-1 Extract on Improving the β_2_-Adrenergic Receptor/PKA Pathway in Bronchial Smooth Muscle

It is known that β_2_-adrenergic receptors (β-AR) in the autonomic nervous system bronchodilation. To determine whether the bronchodilating effect of YG-1 extract occurs through the β-AR, the bronchodilating effect was investigated by pretreatment with propranolol (1 or 100 μM), a non-selective β_2_-adrenergic antagonist. As a result, compared to the relaxation effect of the YG-1 extract, a significant blocking effect of bronchial relaxation was observed at 35.61 ± 11.01% by pretreatment with propranolol at a concentration of 100 μM (Figure 4A). In addition, it is known that smooth muscle induces relaxation by reducing Ca^2+^ levels by converting PKA to cAMP generated by the activity of adenylate cyclase (AC). As a result of confirming the bronchial relaxation effect of the YG-1 extract by pretreatment with KT5720 (10 or 100 μM), a PKA inhibitor, the relaxation effect was reduced to 35.61 ± 11.01% at the 100 M concentration (Figure 4B). As shown in Figure 3 above, YG-1 extract confirmed an increase in cAMP production in bronchial tissues. As a result of confirming whether YG-1 treatment had an effect on cAMP production when PKA blocker was treated, the amount of cAMP production increased by YG-1 treatment was significantly decreased by KT5720 (Figure 4C). Therefore, it is considered that YG-1 extract has a bronchial relaxation effect through the β2-adrenergic receptor/PKA pathway.

### 3.4. Effect of YG-1 on Reducing Bronchial and Lung Fibrosis in PM2.5-Exposed Airway Inflammation Mice

To investigate the inflammatory effects of PM2.5 on the respiratory tract, C57Bl/6 mice were exposed to PM2.5 using a ultrasonic nebulizer aerosol chamber. After the mice were sacrificed, bronchial and lung tissues from all groups were collected and analyzed. The bronchi and lung fibrosis was confirmed using PAS (Figure 5A), masson’s (Figure 5B or Figure 6A) and orcein (Figure 6B) staining. As shown in Figure 5 and Figure 6, significant pulmonary fibrosis was observed in the peribronchial, perivascular, and alveolar spaces of the lungs upon exposure to PM2.5. On the other hand, it was confirmed that fibrosis of the bronchi and lungs was improved by treatment with YG-1 extract.

### 3.5. Effect of YG-1 on Reducing Bronchial and Lung Inflammation in PM2.5-Exposed Airway Inflammation Mice

Histopathological evaluation and pro-inflammatory cytokine levels were evaluated to confirm bronchial and lung inflammation levels. Immunohistochemistry (IHC) staining showed that YG-1 treatment significantly reduced the expression of β-AR, TGF-β, and collagen IV in PM2.5 exposure mice bronchial tissues (Figure 5C–E). As a results, the expression level of β-AR was decreased in the histological evaluation of the bronchial tissues in PM2.5 exposure mice, whereas the expression levels were increased by treatment with YG-1 (Figure 5C). In addition, IHC staining showed that YG-1 treatment significantly reduced the expression of TGF-β and collagen IV in bronchial (Figure 5D,E) and lung (Figure 6C,D) tissues of mice exposed to PM2.5. Furthermore, we investigated whether treatment of YG extract in the airways of PM2.5-exposed mice had an effect on inflammatory cytokines and NLRP3 inflammasome activation-associated protein expression and gene levels. As shown in Figure 5 and Figure 6, PM2.5 exposure to mice increased the expression level of inflammatory cytokines in bronchia (Figure 7A(a,b) and lung tissues (Figure 8A(a,b) compared to the control group. Similarly, higher mRNA and protein levels of TNF-α, IL-1β, and IL-6 were identified in the bronchial (Figure 7B(a–c) and lung tissues (Figure 8B(a–f) of mice treated with PM2.5 (Figure 8A(a,b). Taken together, Inhibition of the NLRP3/caspase-1 pathway by YG-1 alleviated lung inflammation in PM2.5-induced mice model. It was confirmed that the treatment of YG-1 improved the activated NLRP3/caspase-1 pathway in the PM2.5-induced mice model. Thus, YG-1 treatment in mice with lung inflammation caused by PM2.5 exposure has an effect of improving inflammation.

## 4. Discussion

This study was conducted using YG-1 extract mixed with *Lonicera japonica, Arctii Fructus, and Scutellariae Radix*, which are natural products used for respiratory diseases in actual clinical practice. The YG-1 mixed extract was prepared so that natural products could create synergy, and the bronchodilation effect of the YG-1 extract was confirmed. We also evaluated the anti-inflammatory effects of YG-1 in a mouse model of acute bronchial and lung inflammation exposed in PM2.5 exposure mice. 

The two most common bronchodilators used to reverse airway constriction act through stimulation of β_2_-adrenergic receptors (such as salmeterol) or antagonism of muscarinic receptors (such as ipratropium) [23]. In our study, the relaxation effect of YG-1 is mediated through β2-adrenergic receptor stimulation. When β-adrenergic receptor (β_2_-AR) and protein kinase A were blocked by propranolol and KT5720, the bronchodilation effect induced by YG-1 was specifically inhibited. When β_2_-ARand protein kinase A were blocked by propranolol and KT5720 [24], YG-1 induced bronchodilatation was inhibited.

It is generally accepted that stimulation of β_2_-adrenergic receptors increases cyclic adenosine monophosphate (cAMP) levels to mediate airway smooth muscle cell relaxation by activating adenylyl cyclase via the receptor-associated G protein. Respiratory disorders, such as asthma and sore throat, induce contraction of airway smooth muscle cells and airway hyperresponsiveness [25]. To alleviate these acute and chronic airway constrictions, β-adrenergic agonists that relax airway smooth muscle cells are usually administered [24]. The mechanism of action of cAMP is to induce airway smooth muscle cell expansion through stimulation of protein kinase A (PKA) [26,27]. Acts via the cAMP-linked intracellular pathway in airway smooth muscle relaxation, suggesting that it may be an important secondary messenger in bronchodilation [28]. Our study found that increased cAMP production due to YG-1 treatment was significantly reduced by PKA blockers, which resulted in bronchial dilation through beta-AR/PKA pathways. Therefore, YG-1 extract is considered to be of sufficient value as a bronchodilator.

Excessive exposure to PM2.5 gradually adsorbs to the bronchi and lungs and progresses to acute respiratory distress syndrome, eventually requiring bronchodilation for breathing [29,30]. Therefore, YG-1 extract with bronchodilating effect was applied to animal models of acute bronchial and pulmonary inflammation. PM2.5 is a very tiny particle size that can reach almost any organ in the body through blood flow [30]. In particular, the respiratory airway is a tissue that PM2.5 directly affects through respiration. High PM2.5 concentrations in the atmosphere have been reported to increase heart and respiratory diseases [31]. 

Inflammation is a complex pathophysiological process, and it is the expression of a biological defense mechanism against various types of infection or irritants among in vivo metabolites [32]. The main symptoms of the inflammatory reaction are fever, redness, pain, and edema. Nonsteroidal anti-inflammatory drugs are mainly used for the treatment of symptoms, but they are accompanied by various side effects such as gastrointestinal disorders and renal toxicity [33]. Inflammation releases mediators that can induce organ contraction, mucus secretion, and structural changes. TGF-β has been shown to affect many structural cells in vitro and in vivo and implicated in asthma and other inflammatory and immune-mediated lung and bronchial remodeling processes. [34]. We confirmed the increase in airway smooth muscle expression of TGF-1 through the dyeing of bronchial tubes and lung tissue, and confirmed that it was improved by YG-1 extract. TGF-b1 is widely known in many institutions. When structural immune cells and asthma deteriorate, TGF-β1 expression increases in the airway epithelial, which is the main expression area [35,36] Because, PM2.5 has a wide impact on human health [37], it is very important of research to evaluate the improvement effectiveness of YG-1 extracts in PM2.5 inhalation acute lung inflammation-causing mice. Our study used C57BL/6 mice to evaluate the potential mechanism of acute lung inflammation induced by PM2.5 and to confirm the efficacy of YG-1. Additionally, actors involved in the regulation of the proinflammatory cytokines, and IL-1β were also investigated in this model. Exposure to PM2.5 is characterized by the appearance of emphysema and inflammation [38]. In our study, lung histopathology and proinflammatory cytokine levels were detected to assess lung inflammation. As shown in Figure 5 and Figure 6, PM2.5 exposure revealed marked pulmonary inflammation in the peribronchial, perivascular and alveolar spaces of the lung. Previous studies have confirmed that the inflammasome promotes inflammation in a mouse model of PM2.5-induced lung inflammation and that the chronic inflammatory response triggered by various immune cells is important [39,40]. The NOD-like receptor protein 3 (NLRP3) inflammasome is an intracellular multiprotein complex that includes NLRP3, apoptotic speck protein (ASC) and pro-caspase-1 [41]. Pro-interleukin-1β (IL-1β) and pro-IL-18 are converted to mature bioactive forms and released into the extracellular space [36]. IL-1β is a pro-inflammatory cytokine involved as an effector of the NLRP3 inflammasome and is known to increase the incidence of respiratory diseases induced by PM2.5 [42]. As reported in several studies, PM2.5 exposure is known to induce pulmonary inflammation by inducing IL-1β signaling activation [43], and YG-1 extract was found to reduce this in our study. In addition, activation of the NLRP3 inflammasome is known to accelerate pulmonary fibrosis caused by airborne particulate matter [44]. Also, Airway remodeling, one of the main characteristics, shows an increase in airway smooth muscle mass [45]. By contrast, YG extract was confirmed to reduce fibrosis. Various studies have shown that activation of the NLRP3/caspase-1 pathway contributes to the inflammatory response through the onset of diseases such as airway inflammation and chronic obstructive pulmonary disease including pulmonary fibrosis [44]. In addition, it has been reported that TLR4 mainly contributes to the cytokine production induced by PM2.5 [45]. It is known that NLRP3 activates caspase-1 to cleavage pro-IL-1β into mature IL-1β, thereby increasing the expression of inflammatory cytokines to induce inflammation [46,47]. Therefore, in our study, profibrotic pro-inflammatory cytokines such as IL-1β, IL-6, IL-8, and TNF-a [48] were increased in mice exposed to PM2.5. On the other hand, treatment with YG-1 extract decreased the expression of profibrotic cytokines induced by PM2.5 stimulation. Our results suggest that YG-1 extract targeting β-AR signaling in PM2.5-induced airway formation and lung inflammation reduces the production of inflammatory cytokines (IL-6, IL-8, and IL-1β) (Figure 9). Therefore, YG-1 extract is an effective therapeutic strategy for PM2.5-related airway and lung inflammation. 

However, two major limitations of this study currently need to be acknowledged and addressed. First, unfortunately, chemical compounds were analyzed for YG-1 extract, but related studies were not performed. Second, since our results were conducted only on acute lung and bronchial inflammation caused by PM2.5, including the bronchodilatation effect of YG-1, additional studies on chronic diseases are needed.

## 5. Conclusions

In summary, YG-1 extract is considered to have an effect of bronchodilation by increased cAMP levels through the β2-adrenergic receptor/PKA pathway. In addition, PM2.5 exposure was involved in airway remodeling and inflammation, suggesting that YG-1 treatment improves acute bronchial and pulmonary inflammation by inhibited the inflammatory cytokines. Therefore, these results can provide basic data for the development of novel respiratory symptom relievers.

## Figures and Tables

**Figure 1 nutrients-13-03414-f001:**
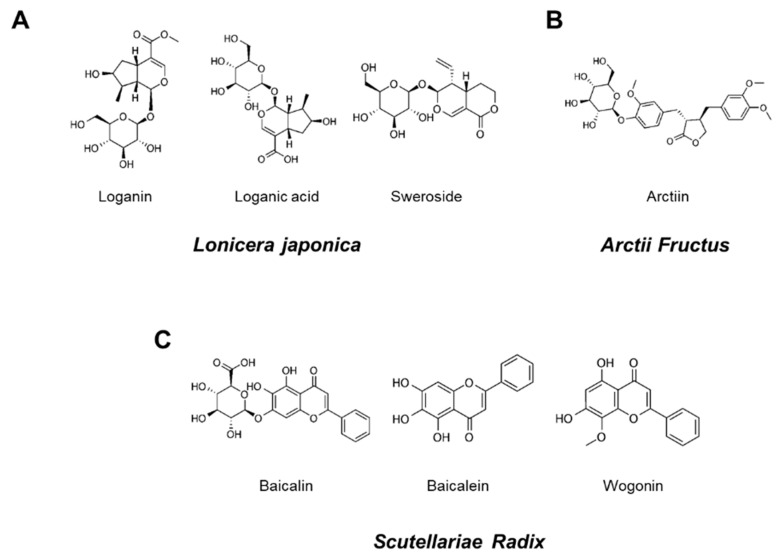
Chemical structures of seven marker components in YG-1 mixed extract. Loganin, logan acid, and sweroside were the main components of *Lonicera Japonica* (**A**). Aarctiin was the main components of *Arctii Fructus* (**B**). Baicalin, baicalein, and wogonin were *Scutellariae Radix* (**C**).

**Figure 2 nutrients-13-03414-f002:**
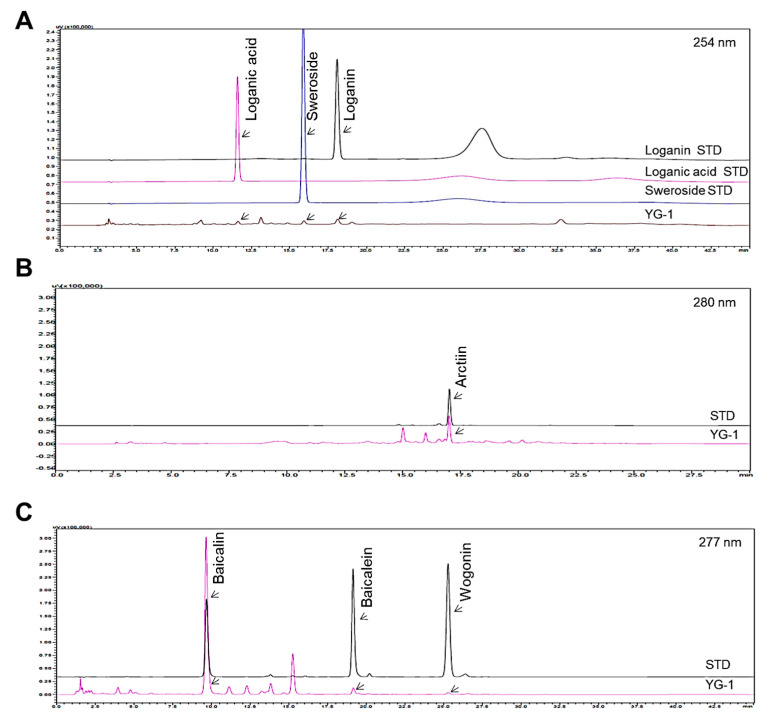
HPLC chromatograms showing peaks corresponding to the marker compounds, loganin, loganic acid, sweoside (**A**), arctiin (**B**), baicalin, baicaein, and wogonin (**C**) of YG-1 extract. HPLC, high performance liquid chromatography.

**Figure 3 nutrients-13-03414-f003:**
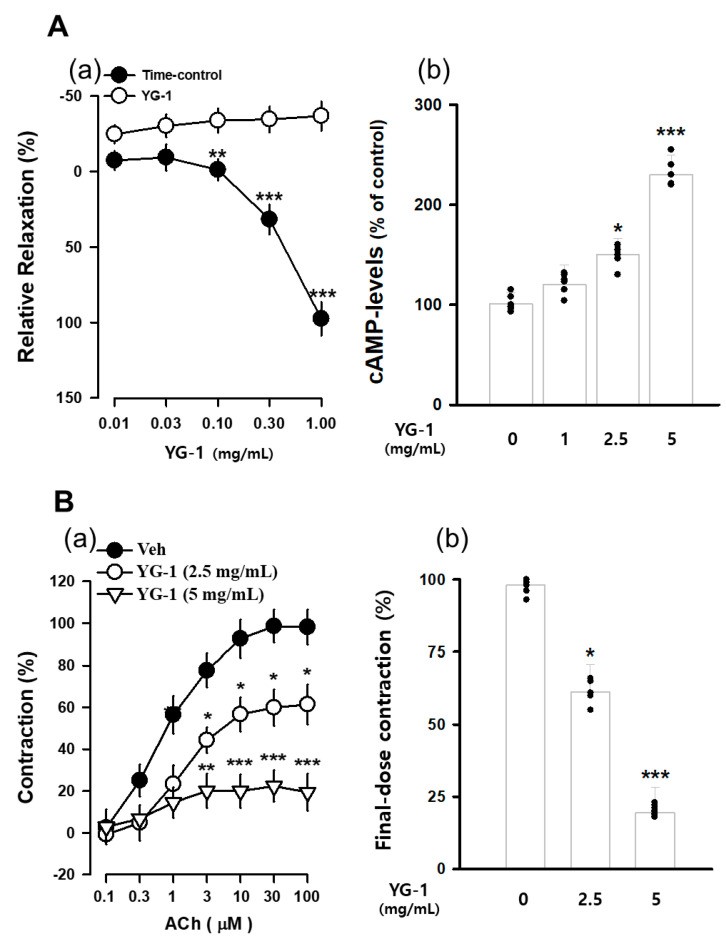
The concentration-accumulation treatment of YG-1 extract shows a dose-response relaxation curve (**A**(**a**)) in the bronchi, as shown in the bar graph (**A**(**b**)). Concentration-dependent bronchoconstriction response curve graph of acetylcholine with and without YG-1 treatment (**B**(**a**)). The bronchoconstriction effect of each group at the highest concentration of YG-1 extract was compared and graphically depicted (**B**(**b**)). Veh, vehicle; cAMP, cyclic adenosine monophosphate. Data are expressed as mean ± standard error. * *p* < 0.05, ** *p* < 0.01, and *** *p* < 0.001 vs. vehicle.

**Figure 4 nutrients-13-03414-f004:**
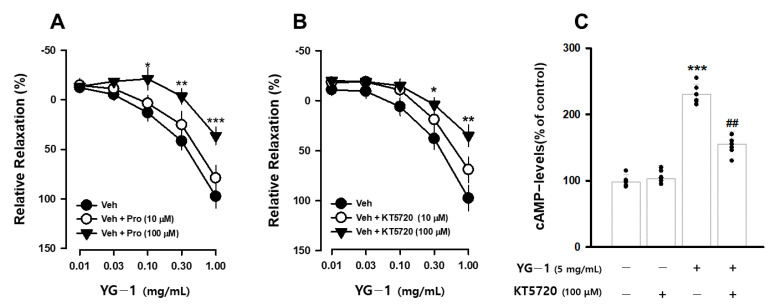
YG-1 extract treatment induced bronchodilation by stimulating β2 adrenergic receptor-mediated PKA activation. Bronchodilation inhibitory effect of YG-1 treatment on acetylcholine (10 μM)-induced bronchoconstriction in rats in the presence of propranolol (10 or 100 μM, **A**) and KT5720 (10 or 100 μM, **B**). The increase in cAMP production by YG-1 in the bronchi was reduced by pretreatment with KT5720 (100 μM. **C**). Veh, vehicle; Pro, propranolol, non-selective β-adrenergic receptor antagonist; KT5720, selective inhibitor of protein kinase A; cAMP, cyclic adenosine monophosphate. Data are expressed as mean ± standard error. *** *p* < 0.001, ** *p* < 0.01, * *p* < 0.05 vs. vehicle; ## *p* < 0.01 vs. YG-1.

**Figure 5 nutrients-13-03414-f005:**
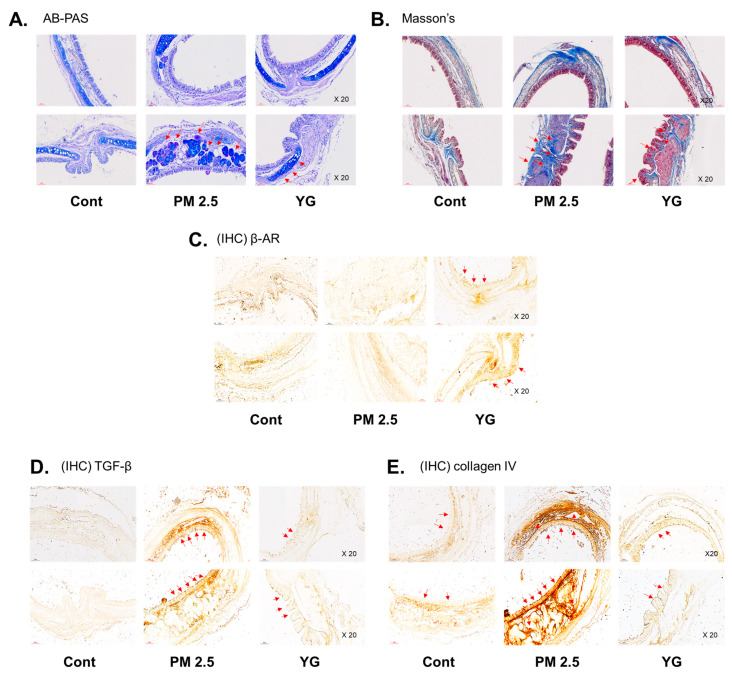
The effect of YG-1 extract treatment on bronchial injury in fine particulate matter (PM2.5)-stimulated mice was histologically confirmed. Representative images of AB-PAS (pseudostratified epithelium, **A**) and Masson’s trichrome (collagen fibers, **B**) stained tracheal in PM2.5 stimulated mice. IHC staining was performed to examine the expression of β-AR (**C**), TGF-β (**D**), and collagen IV (**E**) in bronchial tissues. Red arrows indicate the location of pseudostratified epithelium (**A**) and collagen fibers (**B**); and β-AR, TGF-β (**C**), and collagen IV (**D**) were expressed by immunohistochemistry in tracheal. Histopathological lesions and changes were assessed by histological analyses by optical microscope (magnification ×200; *n* = 3~4 for each group). Cont, control; PM2.5, PM2.5 exposure mice; YG-1, PM2.5 exposure mice + YG-1 treated (200 mg/kg/daily, orally); AB-PAS, alcian blue-periodic acid-Schiff staining; Massnon’s, masson’s trichrome staining; IHC, Immunohistochemistry staining; β-AR, β-adrenergic receptor antagonist; TGF-β, transforming growth factor-β.

**Figure 6 nutrients-13-03414-f006:**
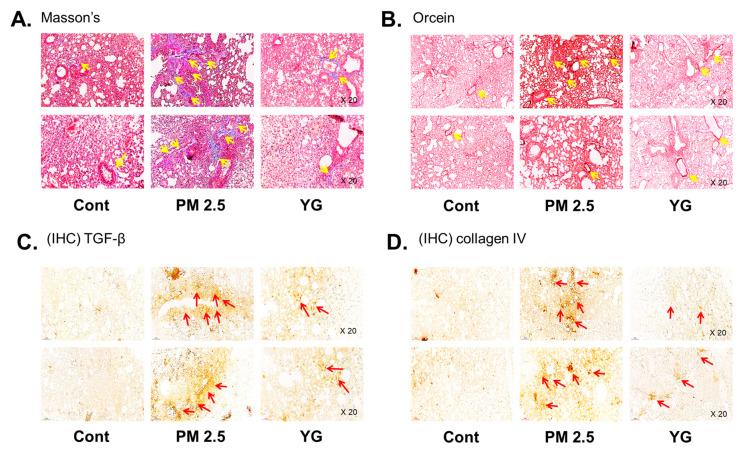
The effect of YG-1 extract treatment on lung injury in fine particulate matter (PM2.5)-stimulated mice was histologically confirmed. To confirm that YG extract treatment inhibited fibrosis of the PM2.5 stimulated mice, histological changes were observed by masson’s trichrome (collagen fibers, **A**) and orcein (elastic fibers, **B**) staining in bronchial tissues. IHC staining was performed to examine the expression of TGF-β (**C**) and collagen IV (**D**) in lung tissues. Yellow arrows indicate the location of collagen fibers (**A**) and elastic fibers (**B**). Red arrows indicated where TGF-b (**C**) and collagen (**D**) were expressed by immunohistochemistry. Histopathological lesions and changes were assessed by histological analyses by optical microscope (magnification, ×200; (*n* = 3~4 for each group). Cont, control; PM2.5, PM2.5 exposure mice; YG-1, PM2.5 exposure mice + YG-1 treated (200 mg/kg/daily, orally); Massnon’s, masson’s trichrome staining; Orcein, orcein staining; IHC, Immunohistochemistry staining; TGF-β, transforming growth factor-β.

**Figure 7 nutrients-13-03414-f007:**
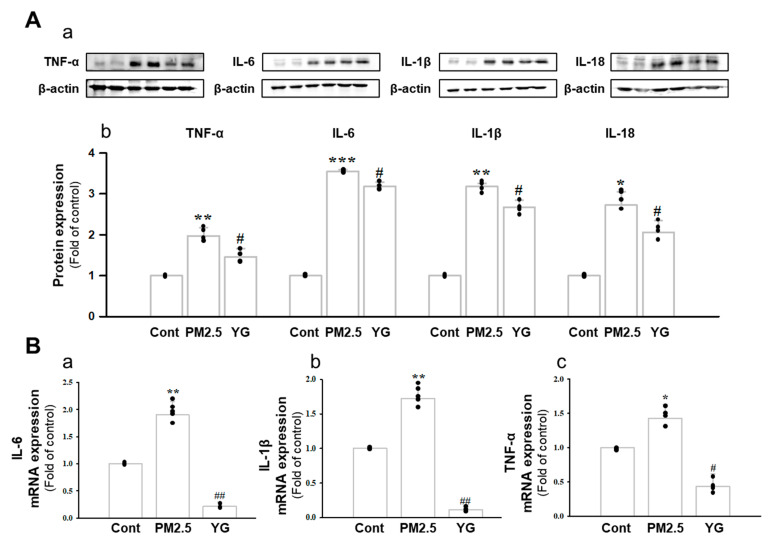
Treatment of YG-1 extract improved acute bronchial and lung injury in fine particulate matter (PM2.5)-stimulated mice. Increased protein expression of TNF-α, IL-6, IL-1β and IL-18 inflammatory cytokines in PM2.5-stimulated mice was improved by YG treatment (**A**). Bronchial damage in PM2.5-stimulated mice increased IL-6, IL-1β and TNF-α mRNA levels and was inhibited by YG-1 treatment (**B**). β-actin was used as loading controls for protein and mRNA expressions (*n* = 3~5 for each group). Cont, control; PM2.5, PM2.5 exposure mice; YG-1, PM2.5 exposure mice + YG-1 treated (200 mg/kg/daily, orally); TNF-α, tumor necrosis factor alpha; IL-6, interleukin 6; IL-1β, interleukin 1 beta; IL-18, interleukin 18. Data are expressed as mean ± standard error. * *p* < 0.05, ** *p* < 0.01, and *** *p* < 0.001 vs. control; # *p* < 0.05, ## *p* < 0.01 vs. PM2.5.

**Figure 8 nutrients-13-03414-f008:**
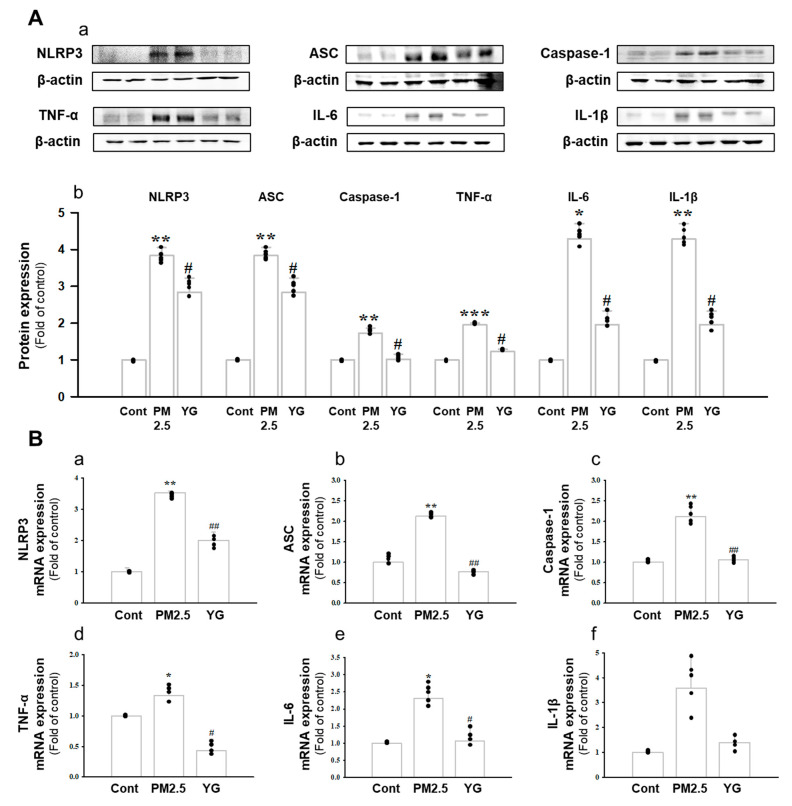
Treatment of YG-1 extract improved lung injury in fine particulate matter (PM2.5)-stimulated mice. Increased protein expression of NLRP-3, ASC, caspase-1, TNF-α, IL-6, and IL-1β inflammatory cytokines in PM2.5 stimulated mice was improved by YG-1 treatment (**A**). Lung injury in PM2.5 stimulated mice increased NLRP-3, ASC, caspase-1, TNF-α, IL-6, and IL-1β mRNA levels and was inhibited by YG-1 treatment (**B**). β-actin was used as loading controls for protein and mRNA expressions. Cont, control; PM2.5, PM2.5 exposure mice; YG-1, PM2.5 exposure mice + YG-1 treated (200 mg/kg/daily, orally); NLRP-3, NOD-like receptor pyrin domain-containning protein 3; ASC, apoptosis-associated speck-like protein containing a C-terminal caspase recruitment domain; TNF-α, tumor necrosis factor alpha; IL-6, interleukin 6; IL-1β, interleukin 1 beta. Data are expressed as mean ± standard error. * *p* < 0.05, ** *p* < 0.01, *** *p* < 0.001 vs. control; # *p* < 0.05, ## *p* < 0.01 vs. PM2.5.

**Figure 9 nutrients-13-03414-f009:**
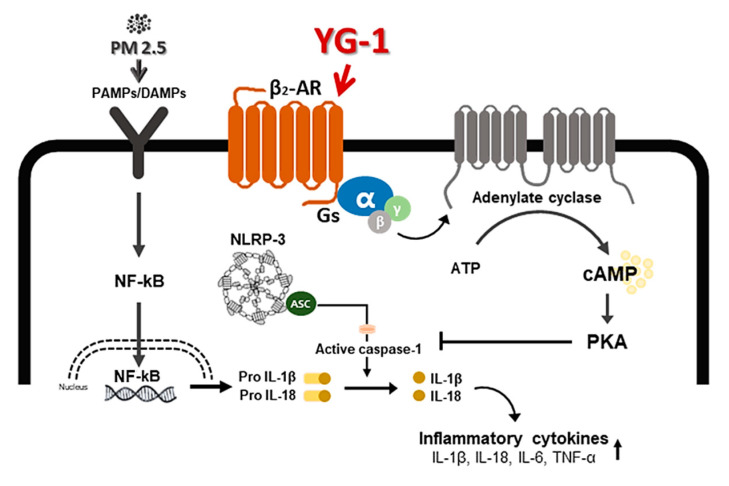
A schematic diagram of the effect of YG-1 extract on airway remodeling in fine particulate matter (PM2.5)-stimulated mice. YG-1 extract improved bronchial and lung inflammation by inhibiting NLRP3/caspase-1 signaling through β_2-_adrenergic receptor stimulation in PM2.5 stimulated mice. Cont, control; PM2.5, fine particulate matter; Gs, G protein; α, G protein alpha subunit; β, G protein beta subunit; γ, G protein gamma subunit; β_2_-AR, β_2_ adrenergic receptor; ATP, adenosine triphosphate; cAMP, cyclic adenosine monophosphate; PKA, protein kinase A; NF-κB, nuclear factor kappa-light-chain-enhancer of activated B cells; Gs, G-protein subtype; NLRP-3, NOD-like receptor pyrin domain-containning protein 3; IL-. 1β, Interleukin 1 beta; IL-18, Interleukin 18; TNF-α, tumor necrosis factor alpha.

**Table 1 nutrients-13-03414-t001:** Mixing ratio of YG-1 extract.

Code	Scientific Name of Source	Ratio	Mixing Ratio
A	*Lonicera japonica,* *Arctii Fructus*	3 1	2
B	*Scutellariae Radix*	2.25	3

## Data Availability

Not applicable.

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
