# Peer review of "YG-1 Extract Improves Acute Pulmonary Inflammation by Inducing Bronchodilation and Inhibiting Inflammatory Cytokines"

_nutrients, 2021, doi:10.3390/nu13103414_

Round 1
Reviewer 1 Report
The ms by Hye Yoom Kim and coworkers describes the YG-1 extract for its bronchodilation effects by increased cAMP levels through the β2-adrenergic receptor/PKA pathway.
I suggest to check the english to avoid some grammar errors as well as to put more emphasis on the conclusions section on the obtained results and possible applications.
Author Response
Reviewer 1
Comments and Suggestions for Authors
The MS by Hye Yoom Kim and coworkers describes the YG-1 extract for its bronchodilation effects by increased cAMP levels through the β2-adrenergic receptor/PKA pathway.
We really thank you very much for your invaluable suggestions and comments.
Comment 1: I suggest to check the english to avoid some grammar errors as well as to put more emphasis on the conclusions section on the obtained results and possible applications.
Response 1: Thank you for reviewing the submitted manuscript. We tried to revise this manuscript grammar overall. Therefore, some fatal grammatical errors have been fixed and marked with red text.
We really thank Reviewer #1 very much indeed.
Reviewer 2 Report
A manuscript „YG-1 extract improves acute pulmonary inflammation by inducing bronchodilation and inhibiting inflammatory cytokines“ is well written and readable. I have few comments.
Figure 3, the legend is a little bit confusing.
Fig. 3Ab, concentration of YG-1 is 0, 1, 2.5, 3, but in the text there is 0, 1, 2.5, 5 mg/ml. which concentration is correct?
Line 264, should be figure 3, instead of fig. 2.
Line 284, correct
Line 399, correct, 2x IL-1β.
From the text, it is not clear, whether the mice were treated before, or after exposure to PM2.5.
There is very little detail in terms of microscopy - how many samples? A number of images per slide? And were the assessments blinded?
Data should be presented as individual data dots, rather than columns
Add limitation of the study.
Author Response
"Please see the attachment"

Reviewer 3 Report
The authors of the manuscript investigated the effect of an extract based on Lonicera japonica, Arctii Fructus, and Scutel lariae radix on bronchodilatation (ex vivo) and acute bronchial and pulmonary inflammation relief (in vivo). The study fits with the scope of Nutrients and should be of interest to the readers of the journal. The methodology used by the authors is robust, and the experiments were carefully done. However, the manuscript is not well written, and the quality of the presentation must be improved. I have the following suggestions.
Major concerns:
- Line 226: it is not clear what component that percentage refers to.
- Line 231: a concentration of the exctract of 5 mg/ml is mentioned, but figures report a maximum concentration of 1 mg/ml or 3 mg/ml. The concentrations tested are not consistent through the manuscript and the figures. Please clarify.
- Lines 238-239: not clear
- Figure 3Aa: I cannot see the dose-dependent effect of the extract.
- The histological observations made by the authors cannot be observed by the reviewer or the reader as the figures are of lousy quality. Please improve them.
Minor points:
- Line 48: the correct acronym for particulate matter is PM and not PM2.5.
- Line 57: it is not necessary to repeat the definition of PM2.5.
- Figure 2 should be moved to results.
- Authors must check the enumeration of the Figures reported in the text.
Author Response
"Please see the attachment"

Round 2
Reviewer 3 Report
I thank the authors for having revised the manuscript according to my comments. I am satisfied with the changes introduced, and I do not have any further comments.